# Barrier Performance of Spray Coated Cellulose Nanofibre Film

**Kirubanandan Shanmugam [1],\*, Narendhar Chandrasekar [2] and Ramachandran Balaji [3]**

1.  Saveetha School of Engineering, SIMATS, Chennai 602107, Tamil Nadu, India
2.  Department of BioNano Technology, Gachon University, Seongnam 13120, Gyeonggi, Republic of Korea
3.  Department of Chemical Engineering and Biotechnology, National Taipei University of Technology, Taipei 10608, Taiwan
\*  Correspondence: kirubanandan.shanmugam@gmail.com

**Abstract:** Cellulose nanofibre (CNF) is the sustainable nanomaterial used for developing high-performance barrier materials that are renewable, recyclable, and biodegradable. The CNF film has very low oxygen permeability; however, its water vapor permeability is significantly higher than that of conventional packaging plastics. The fabrication method influences their barrier properties of the film. A spray-coating CNF on a stainless-steel plate was developed to form a compact film with two unique surfaces, namely a smooth layer on the base side and rough layer on the free side. It improves both the ease of preparation of the film and reduces the water vapour permeability via tailoring the basis weight and thickness of the film through simple adjusting CNF content in the suspension. The air permanence of the film from 1.0 wt.% to 2.0 wt.% CNF suspension is less than 0.003 $\mu$m/Pa·S confirming that is an impermeable film and proves a good packaging material. SEM, optical profilometry, and AFM revealed that the spray-coated surface was smooth and glossy. For sprayed CNF films with basis weight between $86.26 \pm 13.61$ and $155.85 \pm 18.01$ g/m$^2$, WVP were ranged from $6.99 \pm 1.17 \times 10^{-11}$ to $4.19 \pm 1.45 \times 10^{-11}$ g/m·Pa·S. In comparison, the WVP of 100 g/m$^2$ vacuum filtered CNF film was $5.50 \pm 0.84 \times 10^{-11}$ g/m·Pa·S, spray-coated film (of 96.6 g/m$^2$) also show similar permeability at around $5.34 \pm 0.603 \times 10^{-11}$ g/m·Pa·S. The best performance was achieved with spraying of 2.0 wt.% CNF and a water vapour permeability of $3.91 \times 10^{-11}$ g/m·s·Pa. Spray coated CNF film is impermeable against air and water vapour and a potential alternative to synthetic plastics.

**Keywords:** cellulose nanofibre (CNF); spray coating; basis weight; thickness; water vapour transmission rate; water vapour permeability

## 1. Introduction

Packaging materials are mainly composed of synthetic plastics derived from nonrenewable sources. These materials have impeccable barrier performance against oxygen, water vapor, and air; thus, they are conventionally used to protect food and other consumables from physical, chemical, and biological deterioration [1]. Due to an increased awareness in sustainability and circular economy, the renewable materials have been advised for the development of food packaging materials [2]. In addition, renewable materials are ecofriendly and can biodegrade in the environment [3]. Cellulose is the predominant biomaterial available in nature and the main component of woody and nonwoody biomass. Generally, Lignocellulose is biomass consisting of 40% cellulose and is the main feed-stock for cellulose fibres [3]. Paper is a cellulose fibrous packaging, low cost, and sustainable barrier material; however, it does not provide the required barrier performance against air, water vapour, and water owing to the presence of wide pores in the cellulose fibre network [4]. Paper is coated with wax, plastics, or aluminium which are not recyclable and pose a threat to the environment [5]. Therefore, increasing attention has been paid to free-standing nanomaterial films made up of bio-renewable materials to replace synthetic plastics [6].

Recently, cellulose nano-materials such as cellulose nanofibre and nanocrystal exponentially increased in the fabrication of packaging material [7]. The diameter of the cellulose nanomaterial ranges from 5 to 20 nm and provides a large surface area [8]. Owing to the nanosize of the fibres, hydrogen bonds are formed between the fibres and neighbouring fibres, forming a compact network of the nanomaterial. Owing to the compact structure of the fibres, the film has an excellent barrier performance against water vapor, air, and oxygen [2,9].

Cellulose nanofibres from the defibrillation of wood via various methods yield notable properties of cellulose fibres to construct the functional materials of different applications [3]. The reduction in cellulose fibres into either microfibres or nanofibres form a strong fibrous network via hydrogen bonds between the hydroxyl groups between the neighbouring fibres. Therefore, the film made from the cellulose nanofibres has good mechanical properties and higher than that of the nominal cellulose fibres sheet such as paper and paper boards [10]. Cellulose nanofibres consist of two regions namely amorphous and crystalline regions in the fibrous network. These regions create a tortuous pathway for transport of gaseous substances such as air, oxygen, and water vapour. As a result, the barrier performance of the cellulose nanofibre sheet is increased. In addition to that, the dense cellulose nanofibrils network becomes strong via intrafibrillar hydrogen bond and provides good mechanical and barrier properties of the film to replace the synthetic plastics in the conventional practice [8].

Over the past decade, the cellulose nanofibre film/sheet has been of increased interest for preparation, characterization, and their applications in various sectors [5]. Free standing CNF film has been used as barrier material [5], waste water treatment [11], membranes for water [12], membranes for virus removal, substrates for flexible and printed electronics [5,9], adsorbents [11], catalyst, cell culture substrates [9], tissue engineering constructs for soft tissue repair [13], nanocomposites [14], and drug delivery devices [15], etc.

As free standing CNF film has potential in wide applications, it requires a rapid process for the fabrication of the film [12]. The earlier methods are solvent casting, layer by layer assembly, roll to roll coating, and vacuum filtration. Generally, casting was used to fabricate the CNF film at laboratory scale practice. In this approach, time consumption for fabricating the film was more than 24 h and the time to evaporate the water from CNF suspension would be longer and depends on the CNF wt.% solid concentration. The cast film has wrinkles and poor uniformity, which affects the mechanical and barrier properties [11].

Vacuum filtration is a conventional method for fabrication of CNF film and is a bit faster process in the formation of film while comparing with casting [10]. This method has time consumption varying from 4 h to 10 min to achieve 60 g/m$^2$ CNF film [5,9]. Time consumption can be reduced by the addition of poly-electrolytes or increase the CNF solid content above gel point [16]. Gel point is defined as the lowest solid concentration at which a continuous network of fibres exhibits mechanical stability under load and also the transition point from a highly dilute solution of CNF fibres into a concentrated suspension [9]. This approach in the filtration process was capable of reducing the filter resistance [16]. Time for drainage in filtration processes exponentially increased with CNF solid concentration. As a consequence, filtration time was increased with thickness and basis weight of the sheet/film [4,5,10]. In this method, there would be a problem of peeling the film from the filter mesh and handling issues before drying the sheets in the drum drier. The CNF film via filtration sometimes has filter marks, which affects the uniformity of the film. This method has limited capacity in tailoring the thickness and basis of CNF film [9].

Recently, the spraying process is a potential alternative process for fabrication of CNF film/sheet [10]. Spraying CNF suspension on the polished metal surface is an accomplished process for fabrication of smooth CNF film [4,5]. The spraying process is capable of performing coating at higher solid content when compared with vacuum filtration; as such, this process reduces the water content in the suspension for drying the film. The advantages of the spraying process were contour coating and contact-less coating with

substrates, so that the coating is not influenced by the topography of the base surface or substrates. The previous reported work on spraying CNF concludes that the operation time is independent of the CNF solid concentration [4]. The basis weight and thickness of CNF can be tailorable via spraying process [10]. This is why this method has been implemented in the coating of CNF on the paper substrates [17], in the development of free standing smooth CNF films [4], nanocomposites [14,18], coating CNF layer on the membrane for water treatment [18], etc.

This paper describes the barrier potential of spray-coated CNF films and the mechanism of the barrier performance of the film against water vapor. CNF films were compared with conventional plastics and edible polymers have been investigated to evaluate their suitability for packaging applications.

## 2. Materials and Methods

### 2.1. Cellulose Nanofibre

The nomenclature for CNF has not been reported in a consistent manner. Normally, CNF is mentioned as other names such as nanocellulose (NC), microfibrillated cellulose (MFC), cellulose nano-fibrils, cellulose micro-fibrils and nano-fibrillated cellulose (NFC). In this paper, nanocellulose and cellulose nanofibre are commonly used to demonstrate the spray coated film. The CNF from DIACEL, Japan (Celish KY 100S) has been used as a feed stock for the fabrication of CNF film and consists of 25 wt.% solids in the feed stock. The cellulose nanofibrils in Celish KY 100S has a diameter of ~73 nm with mean length of fibre around ~8 μm and an average aspect ratio of $142 \pm 28$. Celish KY 100S was produced by the microfibrilation of cellulose pulp of soft eucalyptus wood with high pressure water. The crystallinity index of Celish KY 100S was evaluated to be 78% [5].

### 2.2. Preparation of CNF Suspension

CNF suspensions were prepared by diluting 25 wt.% Celish KY 100S with de-ionized water and disintegrating for 15,000 revolutions at 3000 rpm in a disintegrator. The dilution was carried out based on the CNF wt.% for spraying to fabricate the CNF film. The CNF solid content for spraying was varied from 1.00 wt.% to 2.00 wt.% of CNF in the suspension. The suspension should be free from fibre aggregates, which blocks the orifice in the spray nozzle. It results in the narrowing down of the spray jet and spray pattern, which affects the formation of the CNF film [10].

### 2.3. Spraying CNF Suspension on the Polished Metal Surface

The CNF suspension of interest was sprayed on the polished stainless-steel surface to fabricate the CNF film. The experimental system to achieve the spraying process is shown in Figure 1. The suspension of interest was sprayed on the steel plate kept on the moving conveyor at a constant velocity of 0.32 cm/s. The used spray system is the professional Wagner system (Model number 117, Wagner, Australia) operating at variable pressure from 100 bar to 200 bar. The elliptical spray jet was formed from the spray nozzle 517 with spray jet angle and beam of 50° and 22.5 cm, respectively. The spray distance between the spray nozzle to the circular steel plate is $30.0 \pm 1.0$ cm. During the spraying process, the pressure driven spray system was allowed to run for 30 s, so that the system can reach equilibrium on forming a uniform spray jet and pattern. It improves the uniformity of the CNF film and the consistency in the thickness and basis weight of the CNF film [10].

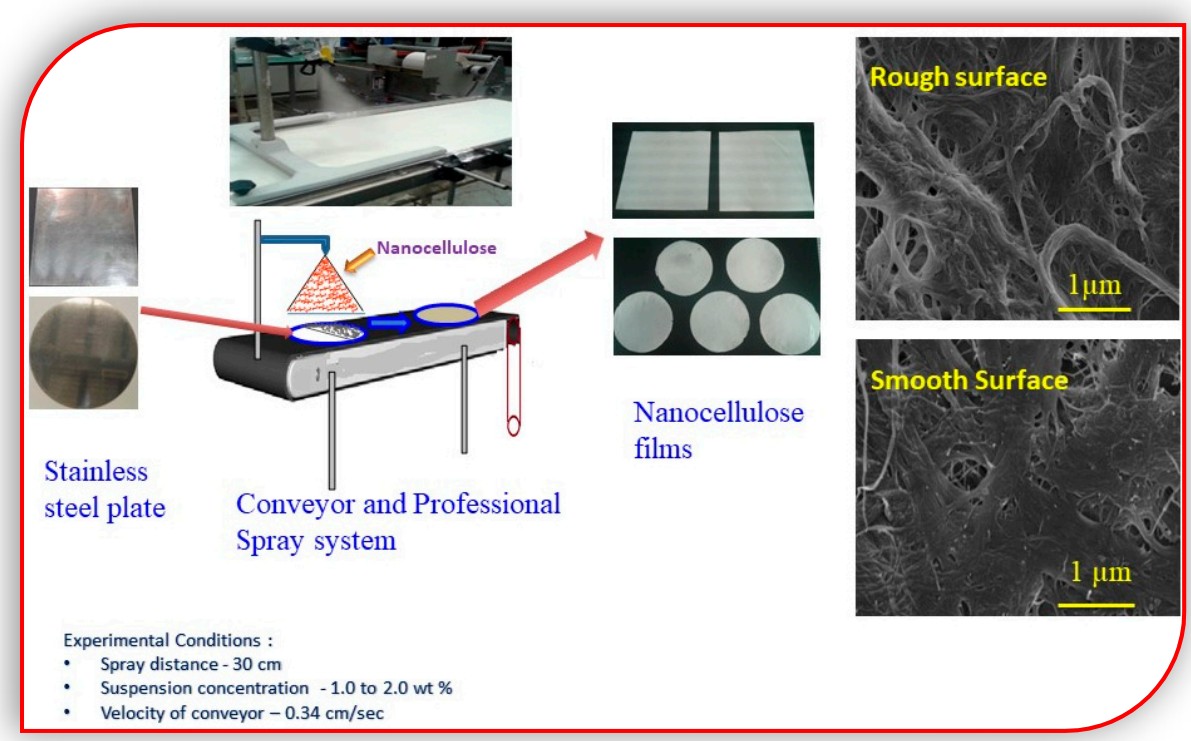

**Figure 1.** Spray coating experimental system.

### 2.4. Drying of CNF Film

The spray coated wet CNF film was allowed to dry in a fume hood where the constant air flow was maintained. Once it is dried in standard laboratory conditions, the film was easily peeled from the stainless-steel plate. The film was conditioned at a temperature of 23 °C and 50% RH for evaluation thickness and basis weight of the film. The film was dried in an air oven maintained at a temperature of 105 °C for 4 h. The basis weight $(g/m^2)$ of the film was evaluated by the ratio between weight of the film and area of the film. The thickness of the film was evaluated via L&W thickness analyser (model no 222, Stockholm, Sweden). The thickness of the CNF film was mapped as per the previously reported method [10].

### 2.5. Evaluation of Air Permeance of CNF Film

The air permeability of the spray coated CNF film was performed with L&W air permanence tester. The tester has an operational range from 0.003 to 100 μm/Pa·S. The mean value of air permeance was evaluated and reported from 3 different areas of each CNF film. The testing procedure was followed as per the standard of The Technical Association of the Pulp and Paper Industry (TAPPI) T 460 [10].

### 2.6. Water Vapour Permeability

Water vapour transfer rate (WVTR) and Water vapour permeability (WVP) were evaluated as per the standard of the American Society of Testing and Materials (ASTM) (E96/E96M-05) method using anhydrous Calcium Chloride ($CaCl_2$). In brief, the CNF film was dried for 24 h at a temperature of 105 °C in an air oven. The required quantity of dried anhydrous $CaCl_2$ was added into the cups and covered by the CNF films. The increase in weight of the cups is caused by the absorption of water vapour by $CaCl_2$ in the cups through the film. The test sample was weighed for every standard interval of time. The variation in weight of the cups with time was noted and the WVTR calculated from the slope of the line between weight and time. The water vapour transmission rate was carried out at 23 °C and 50% relative humidity (RH). The water vapour transmission rate (WVTR)

of CNF film was normalized with thickness of the CNF film and converted into WVP. The mean value of six replicates of each CNF film was reported [5,19].

### 2.7. Surface Morphology and Topography of CNF Film

The surface morphology and topography of the iridium coated CNF film was performed with FEI Novo SEM. The surfaces of the CNF film were evaluated at 1 μm scale bar using the secondary electron (SE) mode-II of FEI Novo SEM. The cross-sectional of CNF film were investigated using Magellan 400 SEM. Furthermore, the surface roughness of both sides of the film at nanoscale was evaluated by atomic force microscopy (JPK Nanowizard 3) and optical profilometry (Bruker Contour GT-I) [4].

## 3. Results and Discussion

Spraying CNF suspension on the polished stainless steel plate is a flexible process for the production of CNF film [4]. The operation time for spraying CNF on the plate is less than 1 min to form 15.9 cm in diameter of the film [10]. In this process, the operation time is independent of CNF suspension concentration. The CNF film via spray coating has two unique surfaces, namely rough surface and smooth surface. The rough surface is a surface exposed to air freely and has very porous and good surface roughness. The smooth surface of the film is shiny and glossy and mimics the part of surface smoothness of the polished stainless-steel plate. This is one of the unique outcome of the spraying process for replicating the polished base surface to the CNF film [10]. Figure 2 shows the spray coated CNF films from the spraying process.

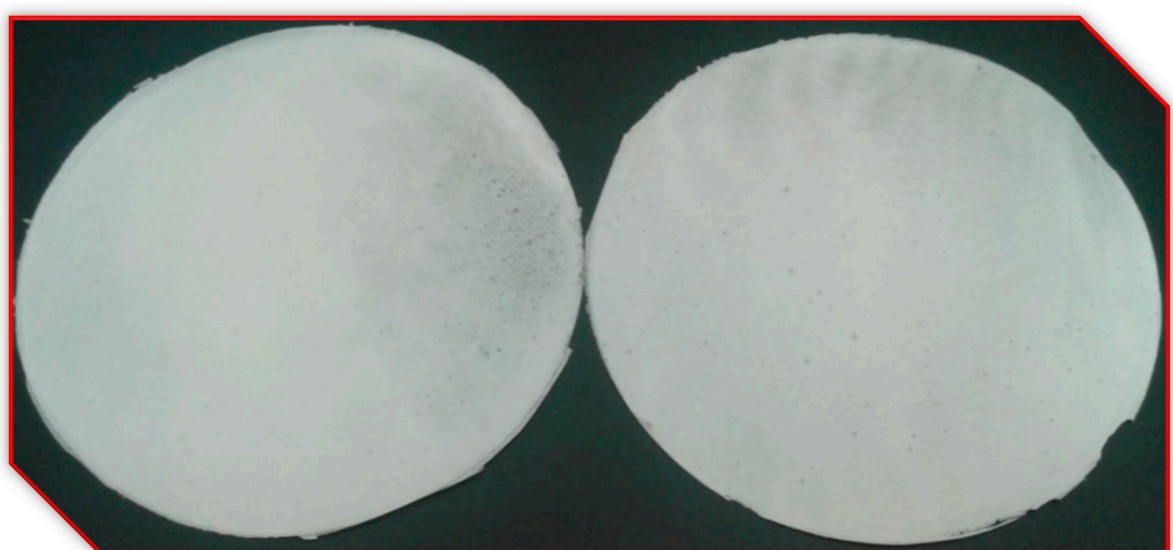

**Figure 2.** Spray coated CNF films.

In this spraying process, the wet film was formed and subjected to drying. The drying of spray coated wet film takes more than 24 h when the drying process is carried out in a fume hood with a constant air flow rate. In the case of drying in an air oven, the removal of water from the wet CNF film will be completed in 60 min. However, the film can be brittle as it is dried at a temperature of 105 °C in an air oven. The spraying process has the potential for being scaled up in industry as the operation time is very minimal. The drying process can be conventionally performed by utilization of waste heat from paper and pulp industry to dry the wet CNF film.

## 4. Thickness Mapping of CNF Film

Figure 3 shows the contour plot of thickness of the spray coated CNF film, fabricated from 1.5 wt.% CNF suspension sprayed on the polished metal plate. The basis weight of

the CNF film is $100.5 \pm 3.4$ g/m$^2$. The thickness plot, read from bottom to top as a conveyor in the experimental setup, moves from one side to the other. The contour plot confirms that the film was very uniform and thicker than that of the film from conventional methods such as casting and vacuum filtration. In addition to that, the variation in thickness distribution of the spray coated CNF film was very minimal and comparable. The mean thickness of the 100.5 g/m$^2$ basis weighed CNF film was reported to be $127.1 \pm 12.1$ $\mu$m [10].

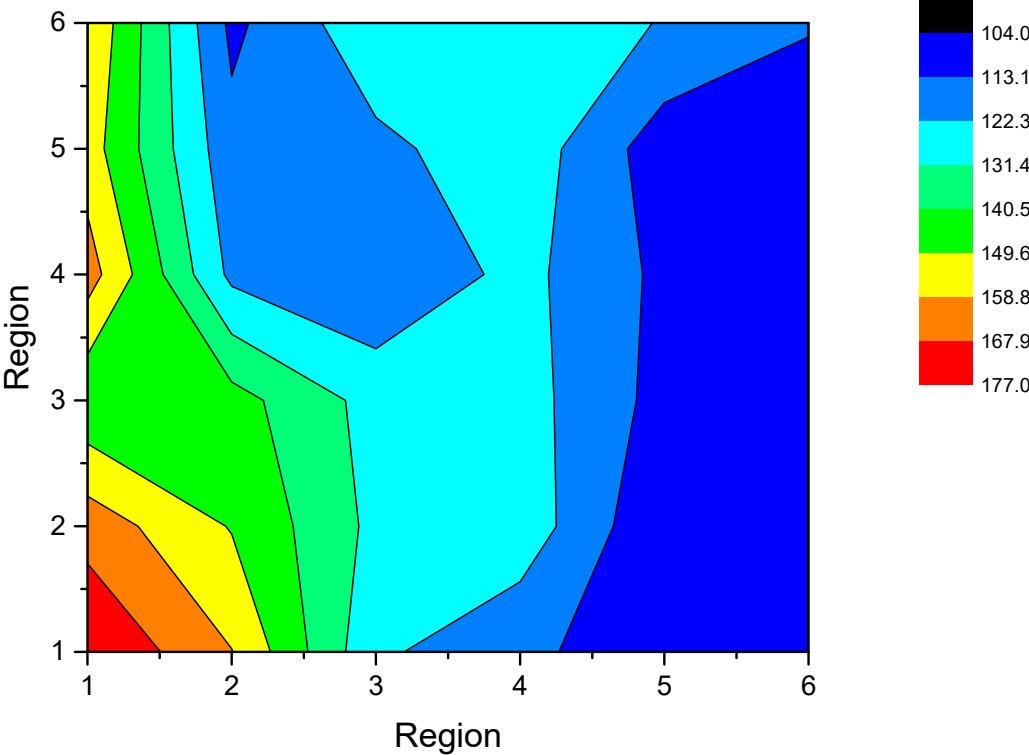

**Figure 3.** Thickness mapping of CNF film.

Figure 4 shows the morphology and topography of the rough side of the spray coated cellulose nanofibre film. The rough side has very porous and had good surface roughness. The rough surface has wider distribution of cellulose nanofibrils and CNF fibre aggregates on the surface of the film. The fibre aggregates were formed by the coalescence of cellulose nanofibres spray jet from the spray gun. In addition to that, the wider distribution of cellulose nanofibrils and cellulose nanoparticles/nanofibre aggregates produce a tortuous pathway for transfer of oxygen, air, and water vapour resulting in a boost of barrier performance of the film. Figure 4 shows the large diameter of fibre aggregates formed via the hydrogen bonds between adjacent fibres and neighbouring cellulose nanofibres. This is why the free side of the film was very rough and there was a high surface roughness due to wide cellulose nanofibre distribution in the film.

Figure 5 shows the smooth surface of the spray coated CNF film, revealing that the film was very compact. The surface was very smooth and glossy and this smoothness was replicated from the surface of a polished stainless steel plate [4]. The cellulose nanofibrils in the film were compressed and gave glossiness to the film and a shiny appearance. The mechanism of the replication of the surface roughness from the base surface to the CNF film remains obscure. The factors controlling the surface roughness of the CNF film were fibre diameter and surface roughness of the base surface. In addition to that, the role of surface roughness of CNF film on barrier performance has so far not been investigated. Spraying the CNF suspension on the polished metal surface produced a smooth film in a single step without any further treatments.

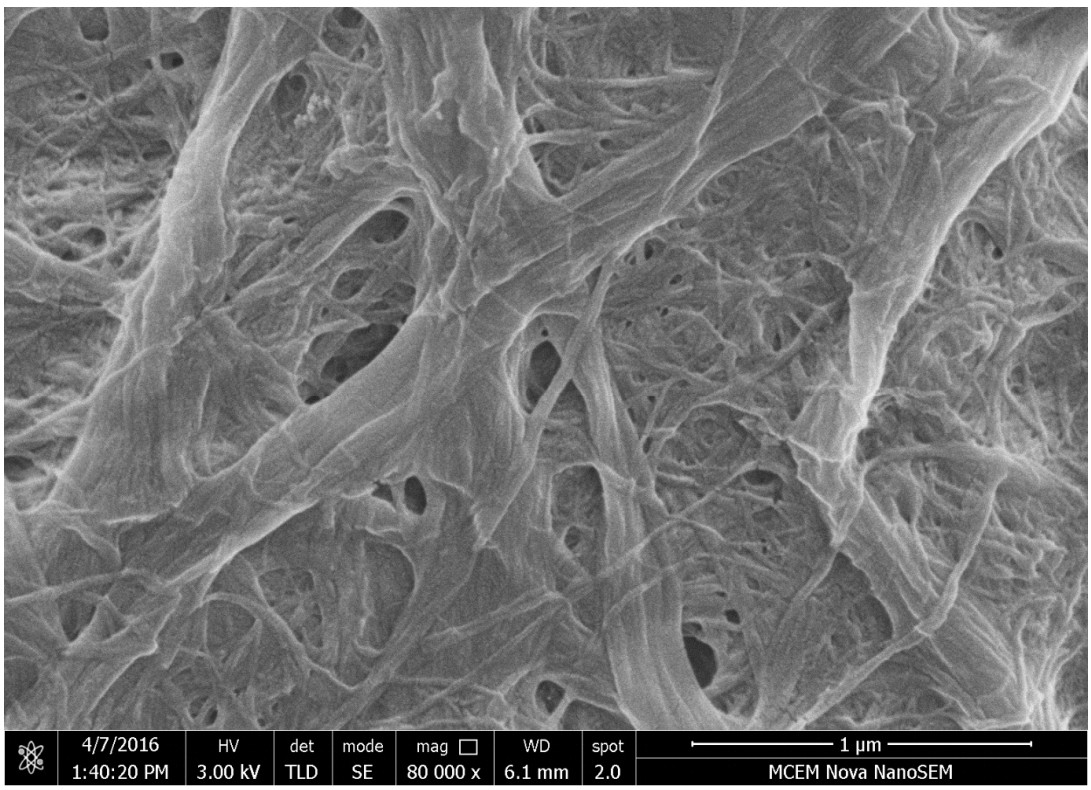

**Figure 4.** Rough surface of the spray coated CNF film.

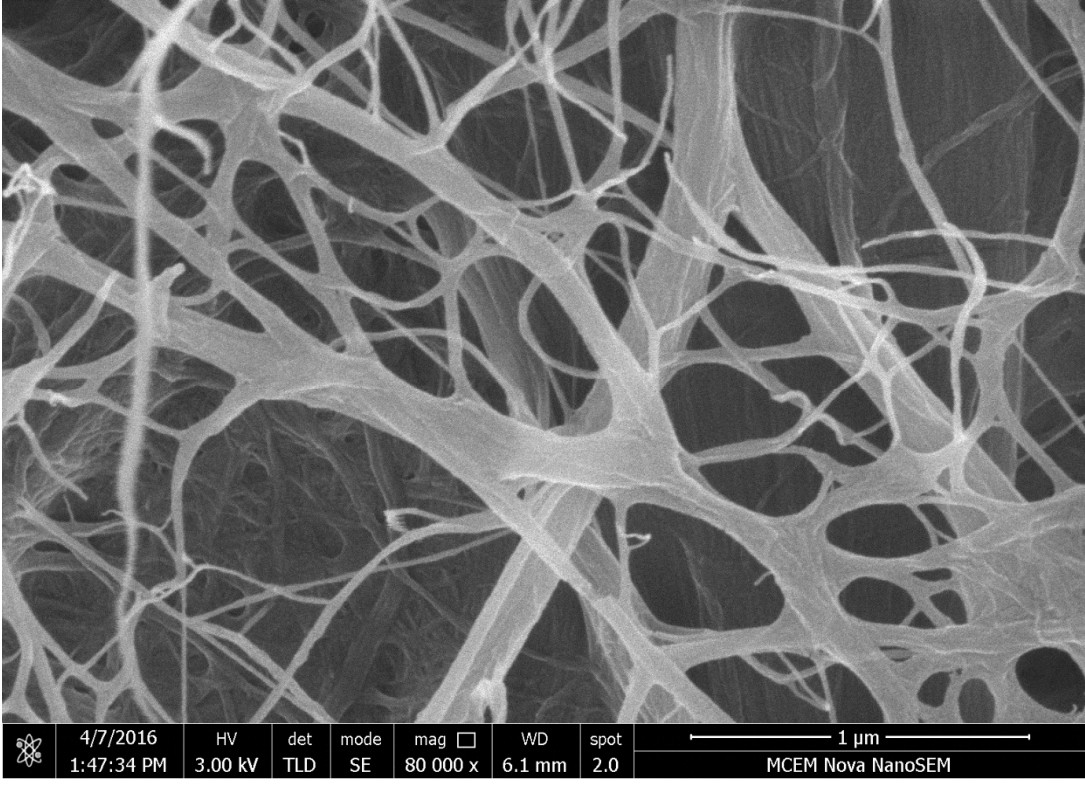

**Figure 5.** Smooth surface of the spray coated CNF film.

*4.1. Cellulose Nanofibre Film from Spraying High Pressure Homogenized Cellulose Nanofibre*

High pressure homogenization is a mechanical process for defibrillation of cellulose nanofibers [5,9]. Homogenization of 1.75 wt.% of CNF was performed to fabricate the

smooth CNF film via spraying. An amount of 1.75 wt.% CNF suspension was fibrillated in a GEA Niro Soavi (Laboratory scale) high-pressure homogenizer and subjected to 1 and 2 passes before spraying. In high pressure homogenization, the number of passes of CNF in the homogenizer controls the diameter of cellulose nanofibrils. The pressure in the homogenizer was maintained at 1000 bar in the first pass followed by 800 bar in the second pass. As a result, the fibre diameter of cellulose nanofibrils reduced from ~70 nm to ~40 nm in the first pass and to ~20 nm in the second pass in the operation of high-pressure homogenization. This method is another approach for reducing the surface roughness of the film [5] and also improves the barrier performance of the film via reducing the pore size [5].

High pressure homogenization is a method of utilizing high pressure energy/efficiency to grind the raw cellulose nanofibre from DIACEL KY 100S, which has a fibre diameter of ~70 nm. The suspension of interest is pumped at a high pressure and passed through a high pressure spring loaded valve assembly in the process equipment of high pressure homogenization. The raw cellulose nanofibre is subjected to a considerable high pressure for fibrillation. The shear force and impact force were produced by the sequence of valve opening and closing in a high pressure homogenizer. These combined forces fibrillate the cellulose microfibres into nanofibers [9].

Figures 6 and 7 show the rough side and smooth side of the spray coated CNF film from homogenized CNF suspension. The rough side has porous and free fibre aggregates and the compactness of the film promotes better barrier performance than the film made from raw cellulose nanofibres. Figure 7 shows the smooth side of the film revealing that the surface is shiny, glossy, and smooth. The film has a complex tortuous pathway for transfer/diffusion of gaseous substances such as oxygen, water vapour, and air. As a result, the barrier performance of the CNF film was improved and was an impermeable film against air passage. The additional SEM micrographs (Figure S1) were added in the supplementary information.

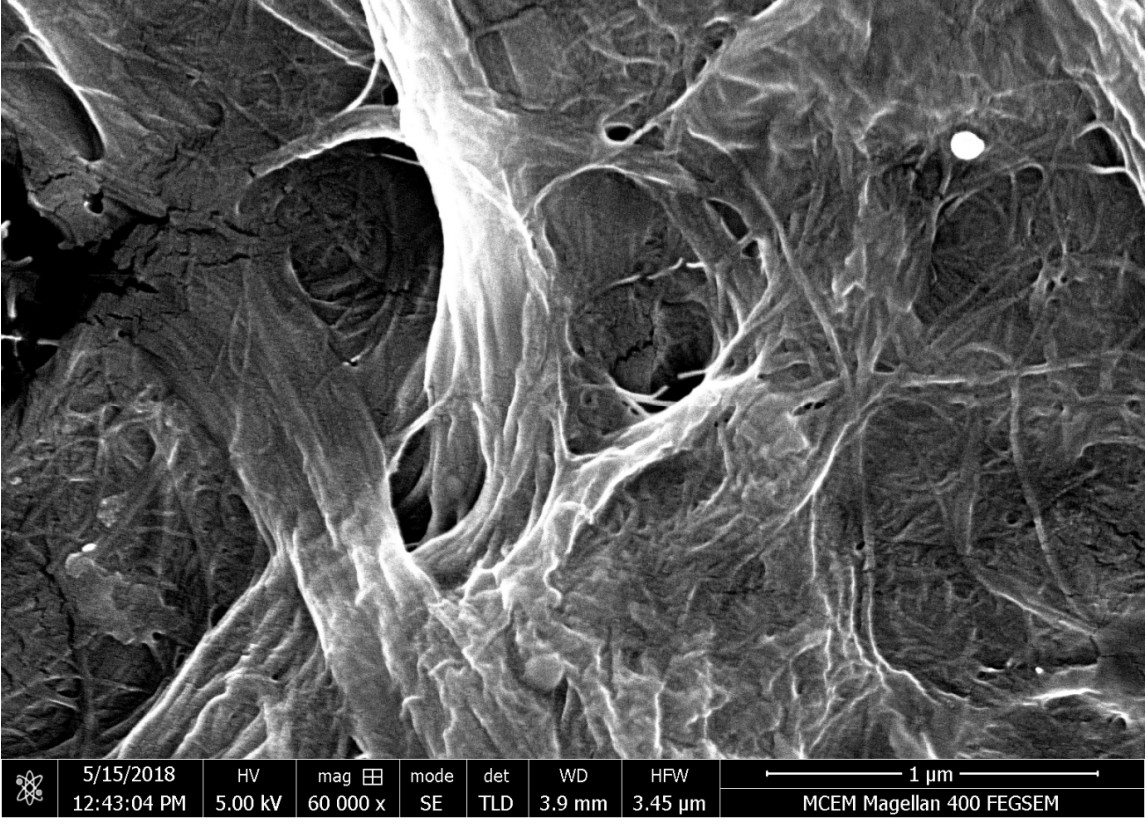

**Figure 6.** Rough side of CNF film via spraying of homogenized CNF.

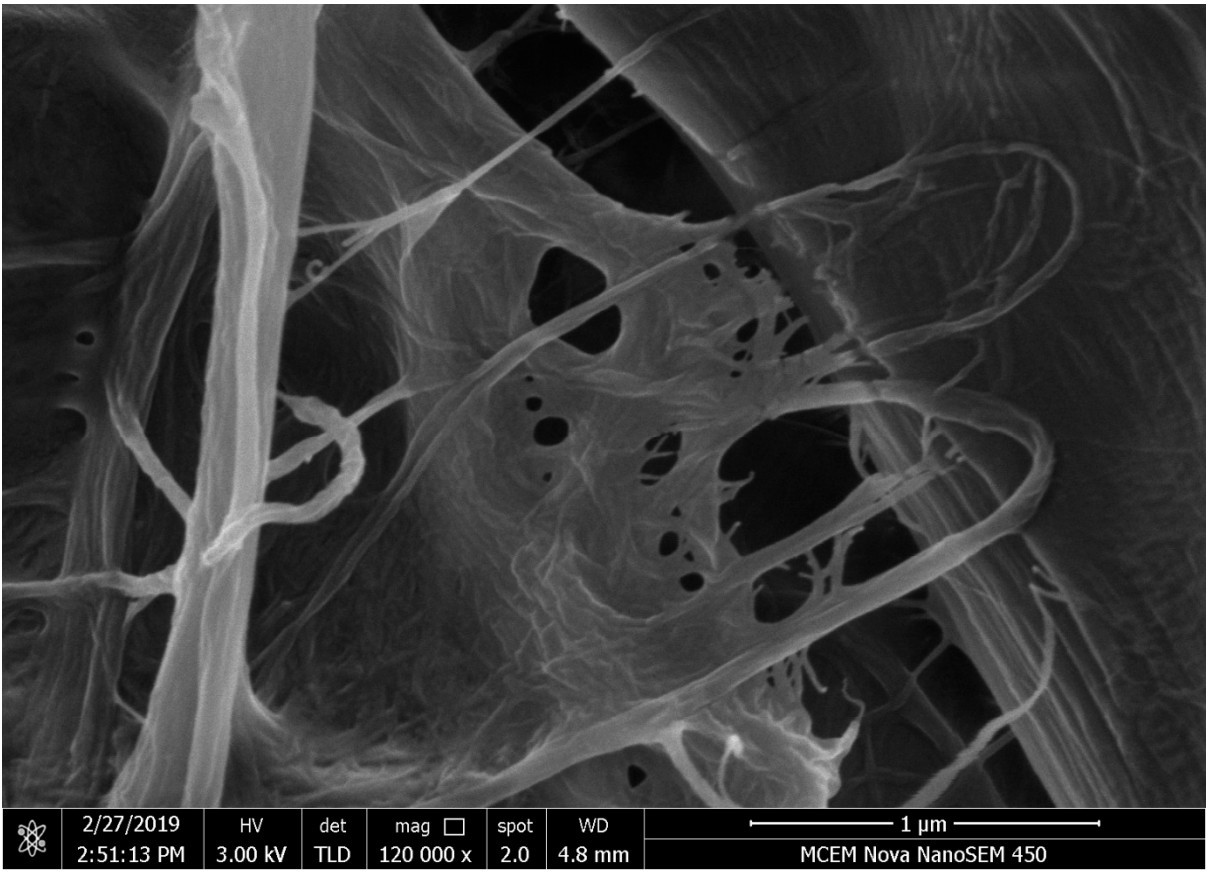

| 2/27/2019 | HV | det | mag ▢ | spot | WD | ⊢—————— 1 µm ——————⊣ |
|---|---|---|---|---|---|---|
| 2:51:13 PM | 3.00 kV | TLD | 120 000 x | 2.0 | 4.8 mm | MCEM Nova NanoSEM 450 |

**Figure 7.** Smooth surface of CNF film via spraying of homogenized CNF.

*4.2. Cross-Sectional Images*

Figures 8 and 9 reveal the cross-sectional images of the spray coated cellulose nanofibre film. The SEM micro graphs confirm that there are many fibrous layers in the film and produce the complex pathway of diffusion of gaseous substances. In other words, the cellulose nanofibre in the film creates a zig-zag pathway for the movement of gaseous molecules.

The optical profilometry images (Figure S2) and AFM micrographs (Figure S3) were added in the supplementary information. The rough side of the film, which has the fibre clumps formed during the preparation of CNF suspension, normally increases the roughness and porosity of the film. The RMS roughness of the CNF film was reported to be 2085 nm on the rough side and to be ~400 nm on the smooth side for 1 cm × 1 cm in section area [12]. The smooth surface shows that the film has uniform smoothness around the film and is shiny and glossy. Without any further treatment, the surface of the film was smooth and had a kind of finished quality of the film via spraying (Shanmugam, Varanasi et al. 2017). The RMS roughness from AFM micrographs of the film was evaluated to be 51.4 nm on the rough side and 16.7 nm on the smooth side for 2 µm × 2 µm inspection area [10].

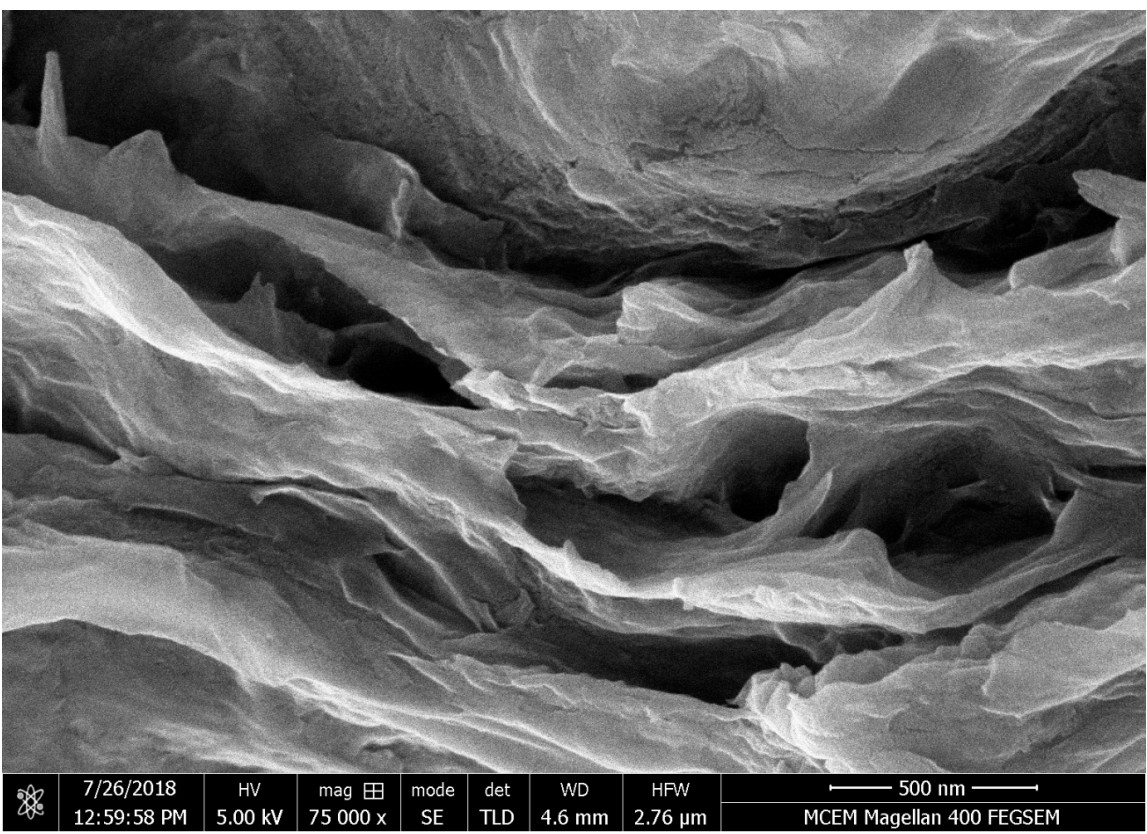

**Figure 8.** Cross-section of spray coated CNF Film from raw cellulose nanofibre.

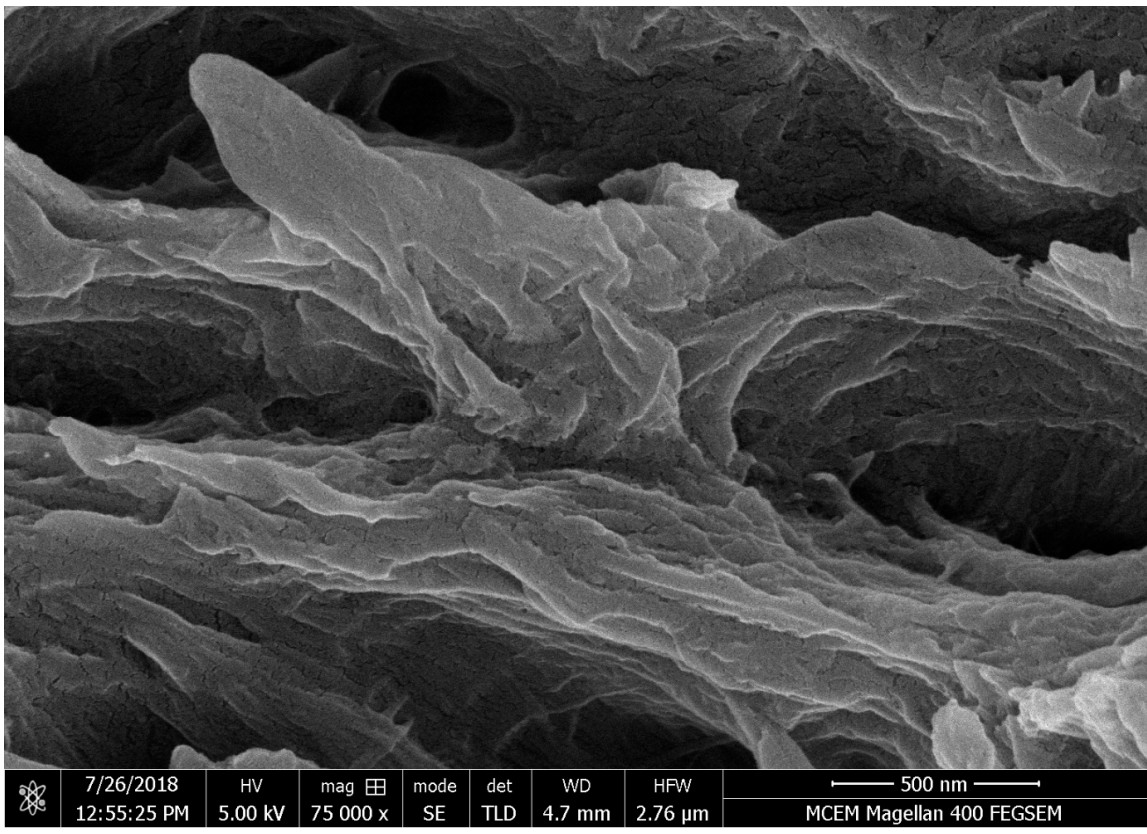

**Figure 9.** Cross-section of spray coated CNF Film from homogenized cellulose nanofibre.

### 4.3. Air Permeability of the CNF Film

Spraying the CNF suspension on the polished stainless-steel plate produced a compact and impermeable film against air and other gaseous substances. The air permeance of the film was reported to be less than 0.0003 microns/Pa·S, confirming that the sheet was impermeable. The air permeance of the cast film varied from 0.009 to 0.013 microns/Pa·s with basis weight of the film from 17 g/m$^2$ to 35 g/m$^2$ [8].

### 4.4. Water Vapour Permeability of CNF Film

Cellulose nanofibre has the ability to form a compact network of cellulose nanofibrils to film as a good barrier material against air and water vapour [8,20]. The film made from cellulose nanofibres has a good oxygen barrier and reasonable water vapour barrier when compared with conventional synthetic plastics [3]. The method is required to tailor the water vapour barrier properties of CNF film via simply adjusting CNF suspension. The resulting film should achieve WVP values comparable with synthetic plastics [5].

### 4.5. Effect of Basis Weight of CNF Film on Water Vapour Transmission Rate and Water Vapour Permeability

Figure 10 reveals the effect of basis weight of CNF film on their water vapour transmission rate (WVTR). It was proved that the relation between basis weight and thickness of the CNF film was linear [10,21]. According to Fick's law of diffusion, permeability is directly proportional to the thickness of the medium [9]. The lower basis weight CNF film was prepared by the spraying of low CNF suspension concentration on the stainless-steel plate. This results in the formation of lower uniformity film with very flimsy elevating of the transport of water vapour molecules. As a consequence, the film has poor barrier performance. The high basis weight film was fabricated via the spraying of high suspension concentration on the polished metal plate. Due to fibre density in high CNF suspension, the thick and good basis weight was formed and the film also has considerable tortuosity for increasing the complex diffusion pathway for water vapour, resulting in good barrier performance of the film [4].

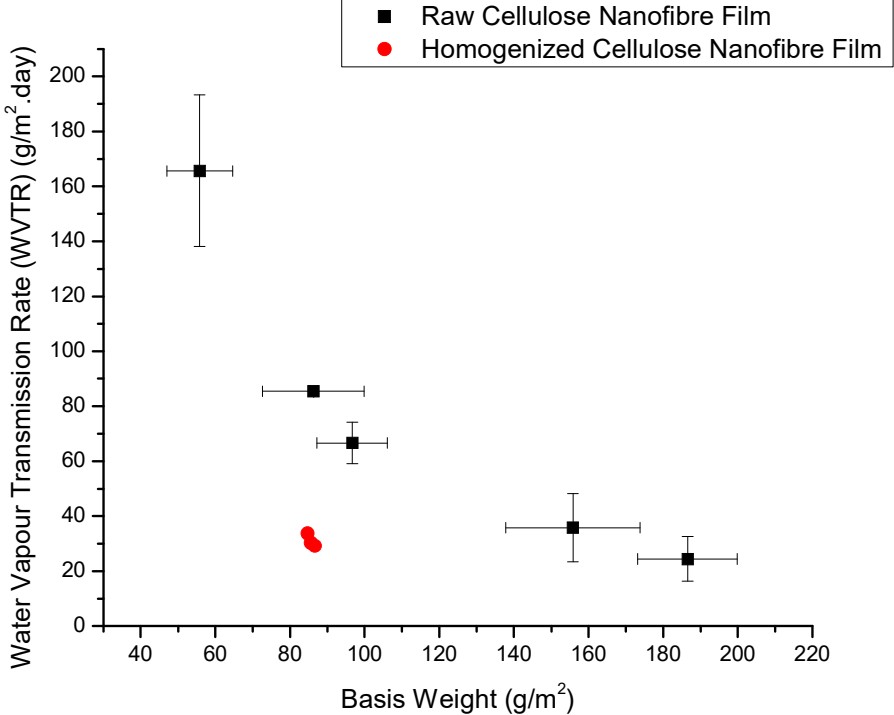

**Figure 10.** Effect of basis weight of CNF film on the water vapour transmission rate (WVTR).

Figure 11 shows the effect of basis weight on the water vapour permeability of the CNF film. The water vapour permeability was evaluated by normalizing the water vapour transmission rate with the thickness of the film [5,19]. The polynomial relationship between WVP and the basis of weight of the film was observed. The red colour in Figures 10 and 11 show the WVTR and WVP of the homogenized CNF film confirming better barrier performance than that of raw CNF film. When cellulose nanofibres were further defibrillated and broken down into nanofibrils, then the film fabricated via spraying of homogenized cellulose nanofibre, produced a good barrier performance due to the reduction in pore size and increased tortuosity complex [2,5].

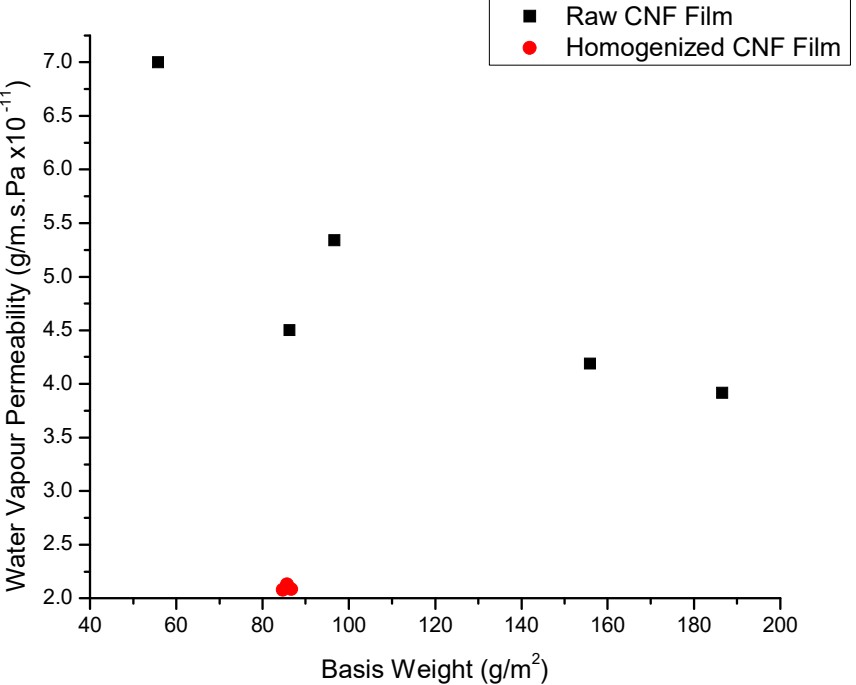

**Figure 11.** Effect of basis weight on the water vapour permeability of CNF film.

*4.6. Effect of Thickness of CNF Film on Water Vapour Transmission Rate and Water Vapour Permeability*

Figures 12 and 13 revel the effect of thickness on WVTR and WVP of the CNF films. The basis weight and thickness relationship is linear and directly proportional to each other [10]. Generally, WVP are normalized with the thickness of the film [19]. Homogenized CNF film has a better water vapour barrier than that of raw CNF film [5].

The effect of fibre diameter on the WVP of CNF film (Figure S4) was added in the supplementary information. It confirms that the diameter of cellulose nanofibrils is one of the strong parameters for tailoring the WVP of the spray coated cellulose films. The diameter of cellulose fibrils is interlinked with pore size and morphology of fibres [5], so that reducing the fibre size results in the lower WVP of the film [5]. Apart from this, the crystallinity in fibres and types of penetrant molecules controls the barrier performance of the film [2]. Temperature and relative humidity are surrounding parameters controlling the WVP of the film [5,19].

Table 1 reveals the water vapor barrier of the CNF film compared to that of other edible barrier materials. The table confirms that CNF is a potential water vapor barrier compared to other edible polymers. However, the conditions and thickness of the film also play a major role in the barrier performance.

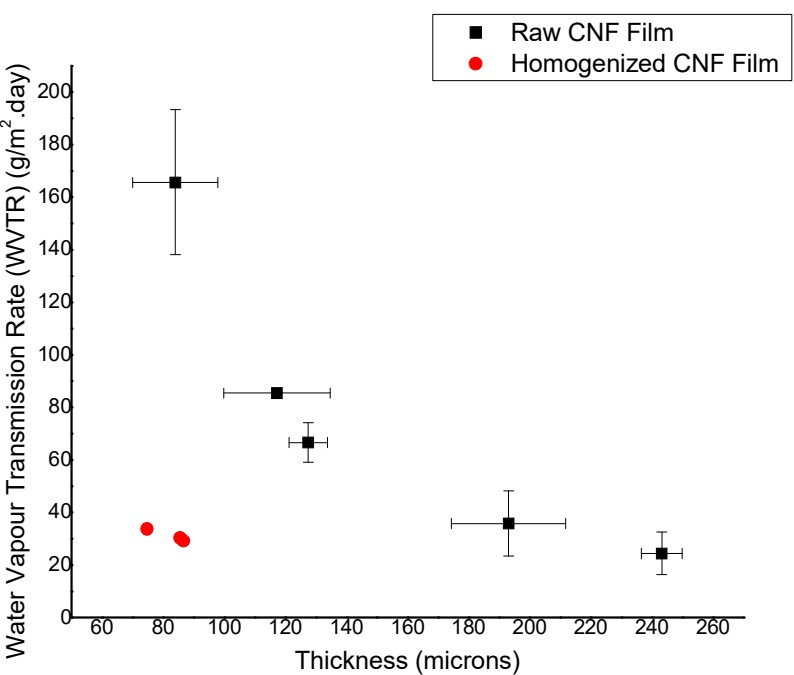

**Figure 12.** Effect of thickness on water vapour transmission rate (WVTR).

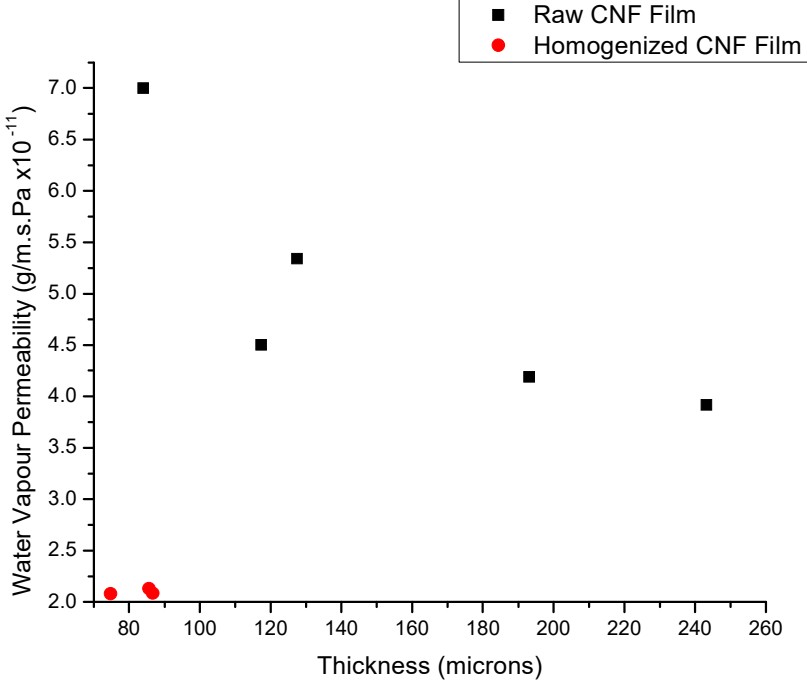

**Figure 13.** Effect of thickness on WVP of the film.

**Table 1.** Comparison of WVP of CNF film with edible polymers film [22].

| Biopolymers | WVP (g/m·s·Pa) | Temperature and RH% |
|---|---|---|
| Cellulose Nanofibre Film | $5.33 \times 10^{-11}$ | 23 °C 50% |
| Ulluco Starch | $4.14–4.84 \times 10^{-11}$ | 25 °C 75–100% |
| Wheat Starch | $130 \times 10^{-11}$ | 25 °C 58–100% |
| Chitosan | $34.5 \times 10^{-11}$ | 25 °C 30–100% |
| Chickpea Flour | $3480–8870 \times 10^{-11}$ | 20 °C 0–100% |

Table 2 confirms the water vapour barrier properties of CNF film comparable with various synthetic plastics. Spray coated CNF film has comparable WVP with other CNF film and cellulose nanofibrils materials. PE is a conventional synthetic plastic packaging material with a WVP of $1.00 \times 10^{-12}$ m·s·Pa, and this value is easily overcome by the WVP of the CNF film $5.33 \times 10^{-11}$ g/m·s·Pa. A difference of one magnitude in the WVP value was observed, and it can be resolved by tailoring the thickness and basis weight of the CNF film. For example, ethylene-vinyl acetate (EVA), polyamide (PA), polycarbonate (PC), low-density polyethylene (LDPE), and polypropylene (PP) have been reported at $3.41 \times 10^{-12}$, $7.54 \times 10^{-12}$, $6.78 \times 10^{-12}$, $8.75 \times 10^{-13}$, and $2.94 \times 10^{-13}$ g/m·s·Pa, respectively [Data taken http://usa.dupontteijinfilms.com, accessed on 21 December 2022]. The difference in the WVP was due to the difference in the thickness of the plastic films. Normally, the thickness of the plastic film varies from 15 to 25 µm, whereas the thickness of the CNF films varies from 60 to 200 µm and depends on the CNF suspension concentration sprayed for film formation. When comparing the value of the WVTR of the CNF film, most of the synthetic plastics were higher than that of the CNF film. This indicates that the CNF film is a potential alternative to synthetic plastics, especially spray-coated CNF films, which are predominant in water vapor barrier performance.

**Table 2.** Comparison of WVP of CNF film with Synthetic Plastics [5].

| Barrier Polymers | Water Transmission Rate (g/m$^2$ Day) | Average Thickness of the Film (µm) | Water Vapour Permeability (g/m·s·Pa) |
|---|---|---|---|
| The CNF Film Spray coated (96.65 g/m$^2$) | $66.62 \pm 7.53$ | 127.5 | $5.33 \times 10^{-11}$ |
| The CNF Film Vacuum Filtration (100 g/m$^2$) | $52.9 \pm 1.2$ | 119.4 | $4.97 \times 10^{-11}$ |
| Recycled Cellulose film from Spray coated NC | 89.5 | $133 \pm 3.92$ | $9.83 \times 10^{-11}$ |
| Cellulose Nano fibrils | 234 | 42 | $8.12 \times 10^{-11}$ |
| Acetylated CNF (Acetylation Time-30 min) | 167 | 46 | $6.35 \times 10^{-11}$ |
| Polyvinylidene Chloride | 3.07 | 12.7 | $1.27 \times 10^{-13}$ |
| Polyethylene (PE) | 16.8 | 18.3 | $1.00 \times 10^{-12}$ |
| Plasticized (PVC) | 118.56 | 12.7 | $4.90 \times 10^{-12}$ |
| Aluminium Foil | 2.376 | 18.3 | $1.42 \times 10^{-13}$ |
| LDPE | 18 | 25 | $8.75 \times 10^{-13}$ |
| HDPE | 9 | 25 | $4.37 \times 10^{-13}$ |
| Oriented Nylon 6 | 260 | 15 | $7.59 \times 10^{-12}$ |
| Oriented Polystyrene | 170 | 25 | $8.27 \times 10^{-12}$ |
| EVA | 70 | 25 | $3.41 \times 10^{-12}$ |
| EVOH | 22–124 | 25 | $1.07 \times 10^{-12}$–$6.032 \times 10^{-12}$ |
| PA | 155 | 25 | $7.54 \times 10^{-12}$ |
| PET | 16–23 | 25 | $7.78 \times 10^{-13}$–$1.12 \times 10^{-12}$ |
| PC | 139.5 | 25 | $6.78 \times 10^{-12}$ |
| PS | 109–155 | 25 | $5.30 \times 10^{-12}$–$7.54 \times 10^{-12}$ |
| PP | 6 | 25 | $2.94 \times 10^{-13}$ |

*4.7. Recommendation for Improving WVP of CNF Film*

The spray-coated CNF film has a compact structure and considerable water vapor barrier compared to CNF films prepared using other methods, synthetic plastics, and biodegradable polymers. To further improve the barrier performance of the CNF film, nano-inorganics were added to the CNF suspension and sprayed on a stainless-steel plate. The resulting nanocomposite acts as a potential barrier against water vapor and air. The barrier properties of these nanocomposites can be tailored by varying the nano-inorganic content of the CNF suspension. Common nano-inorganic materials include bentonite, montmorillonite (MMT) clay, and nanosilica. The common function of the nano-inorganics in the CNF film is to either intercalate or exfoliate in the fibrous matrix and produce a tortuosity pathway for lowering water vapor transmission rate. In addition to that, the mechanical properties of the nanocomposite were improved [5,23]. Another approach for improving the barrier performance of the CNF film is to coat graphene or pullane on the surface of the CNF film to increase the mechanical properties and barrier performance of the film [14]. The incorporation of cellulose nanocrystals/cellulose nanoparticles into CNF suspensions produces a composite with a high-performance barrier against water vapor and other gaseous substances. In situ precipitation of calcium carbonate nanoparticles was deposited into a cellulose nanofibre matrix to fabricate a low water vapor permeability composite via a chemical reaction between sodium carbonate and calcium chloride. The pore volume of the CNF film was reduced by the insertion of calcium-precipitated nanoparticles into the CNF film, resulting in a low permeance of the composite [24].

**5. Conclusions**

The CNF suspension sprayed on the polished stainless-steel plate produced a compact CNF film. The film had two unique surfaces: rough and smooth. The water vapor barrier of the CNF films was tailored by adjusting the CNF suspension from 1.0 wt.% to 2.0 wt.% for spraying. Therefore, the basis weight and thickness of the CNF films were adjusted to tailor the WVP. The WVP of the CNF film is comparable to that of both synthetic plastics and edible polymer films. The operation time for spraying the CNF suspension on the plate to form a 15.9 cm diameter film was less than 1 min. Compared with vacuum filtration, spraying is a rapid process for fabricating CNF films as a water vapor barrier material. This method has the potential to be scaled up for the production of CNF films to replace synthetic plastic materials as barriers against water vapor.

**Supplementary Materials:** The following supporting information can be downloaded at: https://www.mdpi.com/article/10.3390/micro3010014/s1, Figure S1–SEM Micrographs of Rough and Smooth Surfaces of CNF Film; Figure S2–Optifcal Profilometry Images of Rough and Smooth Surfaces of CNF Film; Figure S3–AFM Micrographs of Rough and Smooth Surfaces of CNF Film: Figure S4–Plot between WVP and Fibre Diameter.

**Author Contributions:** Conceptualization, K.S.; methodology, K.S.; software, K.S. and N.C.; validation, K.S., N.C. and R.B.; formal analysis, K.S.; investigation, K.S.; resources, K.S.; data curation, K.S.; writing—original draft preparation, K.S.; writing—review and editing, K.S. and N.C.; visualization, K.S. and R.B. Supervision; Project Administration. All authors have read and agreed to the published version of the manuscript.

**Funding:** This research received no external funding.

**Institutional Review Board Statement:** Not Applicable.

**Informed Consent Statement:** Not Applicable.

**Data Availability Statement:** Data is confidential and unavailable for open access due to privacy and ethical restrictions. The raw data are subjected to various studies in future research.

**Acknowledgments:** K. Shanmugam was thankful to his supervisors Warren Batchelor and Gil Garnier, BioPRIA, Monash University, Clayton, Australia for mentoring his and works on Cellulose nanofibres film fabrication for barrier applications.

**Conflicts of Interest:** The authors declare no conflict of interest.

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
