# Peer review of "Barrier Performance of Spray Coated Cellulose Nanofibre Film"

_2673-8023, doi:10.3390/micro3010014_

Round 1

Reviewer 1 Report

The experimental article “Barrier Characteristics of a Spray-Coated Cellulose Nanofiber Film” corresponds in content to the Micro edition chosen by the authors and is devoted to describing the manufacturing methods and special properties of films obtained from cellulose nanofibers (CNV). Films can be used as packaging material and are preferred over synthetic polymers and plastics due to their biodegradability. The manufacturing technology is not new, but the very idea of manufacturing a multilayer film attracts with engineering ingenuity. The presented material may be of interest to the reader, but for publication it must be reformatted. The strength of the article is the attempt of the authors to illustrate the results of their research in as much detail as possible. But, unfortunately, it was not done correctly enough,

Notes for authors:

1. The main note: it is necessary to fulfill the requirements for articles: the size of the abstract, work with illustrations with the combination of several figures, a list of references with the numbering of sources.

You can use fresh published articles in this edition as a sample, for example

Salanov, A.; Serkova, A.; Zhirnova, A.; Perminova, L.; Kovalenko, G. Supramolecular Aggregation Processes on Carbon Surfaces Occurring in Bovine Serum Albumin Solutions. Micro 2022, 2, 670–678. https://doi.org/10.3390/micro2040045

2. The abstract should be shortened, but reflect the novelty of the author's solution

3. The introduction is presented quite well, but it would be possible to quote articles on this topic from the MDPI publishing house of the last three years.

4. Materials and methods must specify the source of the cellulose (wood, cotton, or other source) and the supplier of the Cellulose nanofiber.

5. Using the example of figure 2 and figure 3: what do these pictures express with similar white circular film samples? Maybe different sides of the samples? Sign additionally in the caption.

6. Please note that there are two figures "Figure 3" in the article.

7. In the process of working on the unification of drawings, the article should be shortened.

8. The following is a list of twelve original works, from which it follows that the topic of making films by the author's method is already well represented:

8.1 SHANMUGAM, K. (2019). Spray coated nanocellulose films-production, characterisation and applications, Monash University.

8.2 Shanmugam, K. (2020). "Preparation of cellulose nanofibre laminates on the paper substrate via vacuum filtration."

8.3 Shanmugam, K. (2021). "Development of Cellulose Nanofibre (CNF) Coating on (1) Metal Surface for Free Standing CNF Film and (2) Paper Substrates for CNF Barrier Laminates." Online Journal of Mechanical Engineering: 10-36.

8.4 Shanmugam, K. (2021). "Preparation of Cellulose Nanofiber (CNF)–Montmorillonite (MMT) Nanocomposite via Spray Coating Process." J. Mater. Sci. Surf. Eng. 8: 978-986.

8.5 Shanmugam, K. (2021). "Spray Coated Cellulose Nanofiber (CNF) Film as an Eco-Friendly Substrate for Flexible and Printed Electronics." Online Journal of Engineering Sciences: 68-81.

8.6 Shanmugam, K., S. Ang, M. Maliha, V. Raghuwanshi, S. Varanasi, G. Garnier and W. Batchelor (2021). "High-performance homogenized and spray coated nanofibrillated cellulose-montmorillonite barriers." Cellulose 28(1): 405-416.

8.7 Shanmugam, K. and C. Browne (2021). Nanocellulose and its composite films: Applications, properties, fabrication methods, and their limitations. Nanoscale Processing, Elsevier: 247-297.

8.8 Shanmugam, K., H. Doosthosseini, S. Varanasi, G. Garnier and W. Batchelor (2018). "Flexible spray coating process for smooth nanocellulose film production." Cellulose 25(3): 1725-1741.

8.9 Shanmugam, K., H. Doosthosseini, S. Varanasi, G. Garnier and W. Batchelor (2019). "Nanocellulose films as air and water vapour barriers: A recyclable and biodegradable alternative to polyolefin packaging." Sustainable Materials and Technologies 22: e00115.

8.10 Shanmugam, K., H. Nadeem, C. Browne, G. Garnier and W. Batchelor (2020). "Engineering surface roughness of nanocellulose film via spraying to produce smooth substrates." Colloids and Surfaces A: Physicochemical and Engineering 44 Aspects 589: 124396.

8.11 Shanmugam, K., S. Varanasi, G. Garnier and W. Batchelor (2017). "Rapid preparation of smooth nanocellulose films using spray coating." Cellulose 24(7): 2669-2676.

8.12 Shanmugam, R., V. Mayakrishnan, R. Kesavan, K. Shanmugam, S. Veeramani, and R. Ilangovan (2022). "Mechanical, barrier, adhesive and antibacterial properties of a pullulan/graphene bionanocomposite coating on sputtered nanocellulose film for food packaging". Journal of Polymers and the Environment 30(5): 1749-1757.

Question: Given your experience in publications, including in Cellulose, present your main results in this article more presentably. At the same time, cite the most significant articles for your new edition.

13. I draw your attention to the fact that the percentage of self-citations is too high.

Author Response

Please find the attachment for Reponses to the reviewer comments

Reviewer 2 Report

Shanmugam et al. developed a method for spray-coating CNF on a stainless steel 16 plate to form a compact film with two unique surfaces, with a smooth layer on the 17 base side and a rough layer on the free side. The best performance was achieved with a spraying of 2.0 wt. % CNF was a water vapor permeability of 3.91 X 10-11 g/m.s.pa. This value indicates that the film is impermeable for packaging applications. Considering the barrier performance of the film, spray-coated CNF film can act as an effective barrier material and a potential alternative to synthetic plastics. This work is interesting for the readers of the Micro. However, the following comments should be addressed before the manuscript can be published.

1.      The abstract and introduction part is too long, and the novelty of this work is unclear; could the authors write concise sentences for this section?

2.      To highlight the novelty of this work, the author should summarise the advantages of using the spray-coating method over other methods, and which properties of cellulose films can be improved by using the spray-coating method compared to other methods?

3.      What are the mechanical properties of sprayed cellulose membranes?

4.      Do room humidity and temperature affect the WVP value of sprayed cellulose films?

5.      What is the water stability of sprayed cellulose membranes as an alternative to synthetic plastics?

6.      There are too many figures included in the manuscript, can the author rearrange the figures by putting similar data/figures together?

Author Response

Please find responses to  the comments from Reviewer 2

Round 2

Reviewer 1 Report

The article has undergone changes, but it could be done even better. It is possible that the authors did not have enough time for this.